# Combinatorial drug repurposing of Valproic acid and Zebularine regulates Krüppel-like factor 4 and β-catenin expression in colon cancer cells

Jeyalakshmi Kandhavelu[1], Kumar Subramanian[1], Natanya Moodley[1], Kasim S. Abass[2], Chandrabose Sureka[3], Meenakshisundaram Kandhavelu[4]*, Paul Ruff[1], Clement Penny[1]*

1 Oncology Division, Department of Internal Medicine, Faculty of Health Sciences, University of the Witwatersrand, Johannesburg, South Africa, 2 College of Veterinary Medicine, Department of Physiology, Biochemistry, and Pharmacology, University of Kirkuk, Kirkuk, Iraq, 3 Department of Basic Medical Sciences, College of Medicine, Prince Sattam Bin Abdulaziz University, Al-Kharj, Saudi Arabia, 4 Molecular Signaling Group, Faculty of Medicine and Health Technology, BioMediTech, Tampere University and Tays Cancer Center, Tampere, Finland

* meenakshisundaram.kandhavelu@tuni.fi (MK); Clement.Penny@wits.ac.za (CP)

## Abstract

Colorectal cancer is the predominant and recurring cancer leading to the second largest cause of mortality, globally. Abnormal regulation of oncogenes and deactivation of tumor suppressor genes over an extended period were key mechanisms underlying colon tumorigenesis. As a result, an effective drug targeting these genes should be explored to combat this disease. We aim to explore the effects of Valproic acid (VPA), a histone deacetylase inhibitor (HDACi), and Zebularine (ZEB), DNA methyltransferase inhibitor, on the expression of the Krüppel-like factor 4 (KLF4) and CTNNB1 (β-catenin) in an early-stage colon cells (SW480 cells) and a late-stage colon cell (DLD-1 cells). Using several in-vitro assays including cell viability, proliferation assay, single cell imaging and analysis and molecular assays including qPCR, protein expression analysis, we have assessed the anticancer properties of both drugs, VPA and ZEB. The synergistic effect of VPA and ZEB could effectively inhibit the proliferation of colon cells than their independent doses. The gene expression profile revealed 2-fold increase in the expression of KLF4 in SW480 cells, with increase to about 22-fold in DLD-1 cells. Notably, CTNNB1 was downregulated than KLF4 in a dose dependent manner, potentially leading to antitumorigenic and anti-proliferative properties of the combinatorial drug. Cellular localization of KLF4 and CTNNB1 protein expression in cytoplasm and nucleus with morphological changes of the cancer cells revealed the onset of programmed cell death. qPCR analysis also showed the downregulation of KLF4 and CTNNB1 upon synergistic activity of ZEB and VPA treated colon cancer cells. Microscopic analysis confirmed the upregulation of KLF4 in ZEB treatment cells leading to DNA methylation. Finally, single cell image analysis has shown the reduced expression of both KLF4 and CTNNB1. Overall, the analysis

**Data availability statement:** All relevant data are within the manuscript and its Supporting Information files.

**Funding:** We would like to acknowledge Natanya Moodley for manual support during the execution of research methodology. The study was supported by funding from the Cancer Association of South Africa (CANSA); and the Medical Research Council (MRC) of South Africa to the Wits/MRC Common Epithelial Cancer Research Centre. JK is grateful to the National Research Foundation (NRF) of South Africa, to the Griffin Trust Fund, University of the Witwatersrand and the Faculty of Health Sciences Research Committee (FRC) for funding. We also would like to thank Department of Internal Medicine, University of the Witwatersrand and all the funding agency.

**Competing interests:** The authors declare no conflict of interest, financial or otherwise.

**Abbreviation:** KLF4, Krüppel-like factor 4; VPA, Valproic acid; ZEB, Zebularine; β-catenin, CTNNB1; DNMTs, DNA methyltransferases; HDACs, Histone deacetylases; CRC, Colo-rectal cancer; HDACi, Histone deacetylase inhibitors; DNMTi, DNA methyltransferase inhibitors; ACTB, β-actin

confirmed that synergistic effect of ZEB and VPA served as a potential anti-colon cancer agent.

## 1. Introduction

Cancer is an important health problem both of the developed and developing world, with high incidence rates and mortality [1]. A recent survey predicts that over 20 million individuals are diagnosed with cancer globally each year [2]. The mortality rates due to Colo-rectal cancer (CRC) are increasing in many economically developing countries, this being mainly due to aging and critical lifestyle factors. Of 19 Sub-Saharan African countries whose colon cancer rates were studied, South Africa and Zimbabwe recorded the highest incidence of CRC [3–5]. Most CRCs develop and grow due to genomic instability, including chromosome instability, DNA repair deficiencies, and aberrant DNA methylation [6]. CRCs are the 2nd most prevalent malignancy globally; nonetheless, therapeutic advancements have halted for many decades [7].

The alteration of oncogenes facilitates the unregulated proliferation of normal cells into malignant cells, as many of these are classified as growth factors and transcription factors, protein kinases, and signal transducers. Prior reports indicate the potential for reversing the tumorigenic features of a cell through the modification of tumor suppressor gene(s) to restore their active function in a specific cell, thereby establishing a model for anticancer therapy.[8]. Krüppel-like factor-4 (KLF4) is considered a potential tumor suppressor gene, expressed in various tissues including the lung and thymus, testis, skin, with the highest expression levels found in the intestinal epithelium [9–11]. The characterization of KLF4 expression in gastrointestinal tissue has been thoroughly conducted. Notably, active KLF4 expression maintains a reduced rate of cell proliferation, stoppage of cell cycle progression and the inhibition of DNA synthesis [9,10]. However, in colorectal cancer cells, KLF4 transcriptional activity is silenced contributing to uncontrolled cell growth. When triggered in vitro, its expression leads to the suppression of DNA synthesis, thereby hindering cell growth [11]. Moreover, it was demonstrated that the interaction with KLF4/β-catenin is crucial for maintaining the homeostasis of the intestinal epithelium and for promoting the development of colon malignancies. β-catenin facilitates the interaction between Wnt signaling and KLF4 leads to blocking the Wnt pathway [12–15].

The two important epigenetic regulators, DNA methylation and histone tail acetylation, both influence patterns of gene expression in an interrelated manner, acting through DNA methyltransferases (DNMTs) and histone deacetylases (HDACs) respectively, to regulate gene expression. Inhibitors of these essential enzymes exhibit anti-tumor properties, capable of affecting several cellular processes, including growth arrest, cellular differentiation, angiogenesis inhibition, cytotoxicity, and apoptosis. These modifications stem from the reinstated acetylation of histones, potentially affecting the expression of downstream genes [16] and DNA methyltransferase inhibitors (DNMTi), which are incorporated into the growing DNA strand and covalently bind DNMTs. This study focuses on Valproic acid (VPA), an HDAC

inhibitor, among many histone deacetylase inhibitors (HDACi) and DNMT inhibitors (DNMTi) that have demonstrated therapeutic enhancements in cellular behaviour. VPA is a class II HDAC inhibitor that promotes cell differentiation in altered hematopoietic cells and various carcinoma cell types, potentially aiding in cancer treatment [17]. VPA is recognized for enhancing β-catenin expression by inhibiting GSK-3β through the phosphorylation of Ser9, hence modulating the Wnt pathway [18]. Zebularine (ZEB), a cytidine analogue, is an important DNMTi characterized by lower toxicity and greater stability compared to 5-azacytidine [19]. As a result of its less toxic nature, this therapeutic agent can also be administered at a reduced dosage for an extended period. Further, it inhibits tumorigenesis and stemness of colorectal cancer [20,21]. Given the distinct epigenetic mechanisms of VPA and ZEB, we propose that their combinatorial repurposing may synergistically enhance therapeutic efficacy in the treatment of colon cancer

In this study, we employed VPA at 1.5 and 5 mM and ZEB at 0.1 and 0.3 mM, doses selected based on prior in-vitro studies demonstrating biologically relevant epigenetic and anticancer effects. The concentrations of VPA and ZEB were selected based on prior in-vitro studies demonstrating epigenetic and anticancer activity. VPA is commonly used at millimolar concentrations in colorectal and other cancer cell lines, with reported $IC_{50}$ values of 2–10 mM, supporting the use of 1.5–5 mM to induce HDAC inhibition without immediate nonspecific cytotoxicity [17,22,23]. However, these concentrations exceed clinically achievable plasma levels (≈0.3–0.7 mM) [24]. Sub- to near-$IC_{50}$ doses of VPA have also shown synergistic effects with other epigenetic agents [25]. Although fewer studies have examined zebularine in colorectal cancer models, zebularine has been shown to induce DNA demethylation and growth inhibition at 100–400 µM (0.1–0.4 mM), particularly in combination with HDAC inhibitors [26,27]. Consistently, zebularine at 50–200 µM produces dose-dependent effects in breast cancer cells, with $IC_{50}$ values of ~88–426 µM [27]. In this study, we evaluated the demethylating and cytotoxic effects of ZEB in early-stage and late-stage CRC cells, explored its potential in combination with other epigenetic modifiers, and assessed its impact on the mRNA and protein expression of KLF4 and CTNNB1.

To test our hypothesis, here we examined at how VPA and ZEB affected the expression of KLF4 in CRC cells at both early and late stages. Subsequently, the combined treatment of CRC cells with DNMTi and HDACi was assessed in relation to the gene expression of KLF4 and the Wnt pathway (β-catenin) in CRC cells. The MCF-7 breast cancer cells were used as a positive control [28], as they have low endogenous KLF4 expression [29].

## 2. Materials and methods

### 2.1 Cell culture and reagents

Two colorectal cancer lines, SW480 (Dukes' type B), DLD-1 (Dukes' type C); and the breast cancer cell line MCF-7 (Health Science Research Resources Bank, Japan) were used to testing the selected drugs. All the cell line were maintained at 37°C, 5% $CO_2$, 95% humidity and cultured in Dulbecco's Modified Eagle Medium/Nutrient Mixture F-12 medium (DMEM/F-12, Invitrogen, Walthem, USA) containing 10% foetal bovine serum (FBS, Invitrogen) and 2 µL/ml penicillin/streptomycin (Lonza BioWhittaker®).

The drugs Valproic acid (98%, P4543) and Zebularine (≥98%, Z4775) were purchased from Sigma-Aldrich, Johannesburg, South Africa. These drugs were dissolved in dimethyl sulphoxide (DMSO, Sigma-Aldrich) to obtain a stock solution; and DMSO was used as a vehicle control. Intermediate dilutions were prepared from the stock solution. In addition, we also used following antibodies for staining: KLF4 Rabbit polyclonal IgG (Santa Cruz, sc-20691, 1:100 dilution), β-catenin Mouse IgG (BD Transduction Laboratories, 610153, 1:100 dilution), Alexa Fluor® 488 Donkey Anti-Rabbit Life Technologies, A10042,1:200 dilution), Alexa Fluor® 568 Donkey Anti-Mouse Life Technologies, A10037, 1:200 dilution).

### 2.2. Cell viability assay

The cells were cultured in 6 well culture plates and grown until the cell monolayer was confluent. In order to achieve cell synchronization, the cells were first serum-starved overnight and treated with 1.5mM and 5mM of valproic acid (VPA) and

0.1mM and 0.3mM of Zebularine (ZEB), for 24 hours. Untreated cells were maintained and cultured as described above. As a vehicle control, 10% FBS/DMEM: F12 with an equivalent amount of the diluent, DMSO replacing ZEB, were also maintained. A set of cells was subjected to check the synergistic effect of VPA and ZEB, designated as SYN1 (1.5mM and 0.1mM ZEB) and SYN2 (5mM VPA and 0.3mM ZEB), for a duration of 24 hours. After the treatment, cells were harvested and pelleted and used for a Trypan Blue cell viability assay [30]. Live and dead cells were counted using a Bio-Rad TC10TM automated cell counter, USA. Cell viability assays were performed by mixing equal volume of TC10 Trypan Blue with the cell suspension from which 10 µL was added to the chamber of the TC10 counting slide for counting.

## 2.3. Real-Time PCR

The mRNA levels of CTNNB1 (β-catenin), KLF4 and ACTB (β-actin) were analysed by RT-PCR. For this, total RNA was extracted from SW480, DLD-1 and MCF-7 cells using the RNeasy Plus Mini kit (Qiagen) according to the manufacturer's instructions. RNA was quantified spectrophotometrically (Nanodrop ND-1000). One hundred ng of total RNA was used in the reverse transcription reaction (ABI Reverse Transcription Reagents kit) The PCR amplifications were performed in a 10µl total volume using SYBR Green PCR master mix (Applied Biosystems) and 3 µL cDNA as a template, together with the primers for each of CTNNB1, KLF4 and ACTB (β-actin), as listed in Table 1. The β-actin (ACTB) mRNA levels were used as internal controls. Cycling conditions were as follows: initial enzyme activation at 94°C for 10 min, followed by 35 cycles of denaturation at 94°C for 1 min; and then annealing and extension at 60–64°C for 1 min. The target gene expression of KLF4 and β-catenin was quantified using qRT-PCR, normalized to the housekeeping gene β-actin, and calculated using the $2^{-\Delta\Delta Ct}$ method, with results expressed as fold changes relative to the untreated control. All qPCR experiments were performed with biological and technical repeats.

## 2.4. Immunofluorescence

The SW480, DLD-1 and MCF-7 cells were seeded onto a sterile glass coverslip and maintained in cell culture condition at 37°C with 5% $CO_2$ for 24 hrs. After reaching 50–70% of confluence the cells were fixed in 3% formaldehyde in PBS for 10 minutes at room temperature. The coverslips were rinsed thrice with PBS, further blocked in 0.5% v/v of BSA/PBS for 5–10 minutes. After this, cells were incubated with the primary antibodies diluted 1:100 in PBS/BSA (5%) for KLF4 and β-catenin, respectively. Following 24 hours of incubation at 4°C in a humidified container lined with damp filter paper, the coverslips were washed with PBS/BSA (5%) to remove any residual primary antibody. This was further incubated with fluorescently tagged secondary antibodies diluted in the ratio of 1:200 in PBS/BSA (5%) and incubated further for 1 hour at 25° C in the dark condition. After washing, the F-actin cytoskeleton and cell nuclei were stained with the dilution at 1:20 ratio using Alexa Fluor® 488 Phalloidin conjugate and 1:10 000 dilution of DAPI, respectively. Coverslips were subsequently positioned on the microscopic slides using Gel MountTM aqueous mounting media (Sigma-Aldrich) and allowed to set for 1–2 hours.

## 2.5. Microscopy

Microscopy was performed using an Olympus IX71 fluorescence microscope with a 60X objective. The higher magnification images were taken by 63X objective of Zeiss Laser Scanning Confocal Microscope 780 and Zen Blue software was used to capture the image. Lasers 405 and 488 nM were used to obtain fluorescence images.

**Table 1. Primers used in Real-Time PCR experiments.**

| PRIMER | UPSTREAM (5'-3') | DOWNSTREAM (5'-3') |
|---|---|---|
| β-actin (ACTB) | GCACACTTAGCCTTCTTTGCACTTTCT | AGGCGTACAGGGATAGCACAGCCTGG ATAG |
| KLF4 | ACCAGGCACTACCGTAAACACA | GGTCCGACCTGGAAAA TGCT |
| CTNNB1 | GCTGGGACCTTGCATAACCTT | ATTTTCACCAGGGCA GGAATG |

## 2.6. Single cell gene expression analysis

To measure the single cell gene expression (KLF4 and CTNNB1), cell image analysis software CellProfiler was used [31] for automated segmentation and data extraction from each selected region of interest. For segmentation, each nucleus was first detected from the blue channel and then the border of cell was propagated into the red channel image. When required, images were smoothed for more accurate cell border detection. The autofluorescence of each image is represented by the mean intensity of the background of the image. Background was detected using two-class Otsu thresholding [32]. Red fluorescent protein signal was extracted from each individually segmented cell. The expression level of either KLF4 or CTNNB1 in an individual cell was described by the mean fluorescence intensity of the detected cell, with the autofluorescence subtracted.

## 2.7. Data analysis and statistics

All the experiments were performed with group size of n ≥ 5 independent samples. The data were represented as mean ± standard error. The experimental outcomes of VPA and ZEB were analyzed for synergism or antagonism by the Chou-Talalay combination index approach utilizing CompuSyn software. Dunnett's multiple comparison statistical test was performed to measure the statistical significance between the tested groups; and $p < 0.05$ was considered as a statistically significant data.

## 3. Results

### 3.1. VPA and ZEB reduce cell viability in a dose-dependent manner

The effects of individual VPA and ZEB treatments, at low and high concentrations, were assessed on the cell viability of the SW480, DLD-1 and MCF-7 cells, respectively. It was observed that the cell viability was reduced more in the DLD-1 cells than SW480 cells (Fig 1A and 1B), with the effects of VPA and ZEB both being dose dependent. The KLF4 over-expressed MCF-7 cell line was used positive control, exhibited a higher susceptibility to the drugs than the colon cancer cells. Cells treated with a combination of 5 mM VPA and 0.3 mM of ZEB showed enhanced cell death in both SW480 and DLD-1 cells (Fig 1C). These results indicated that the combination of the HDACi, VPA and DNA methyltransferase inhibitor, ZEB, inhibit CRC cells more effectively than individually.

We also analyzed the Combination Index (CI) to determine whether the combined effect of the drugs is synergistic, additive, or antagonistic. The CI is calculated based on the dose-response curves of the individual drugs and their combinations. The obtained CI from the two combinations of VPA and ZEB (1.5 + 0.1 & 5 + 0.3) are the most suitable CI values since both are less than 1. Other hypothetical values were used to drive simulation curve in CI plot and to compute the possible CI values and to draw a comparison among the experimental and computational values. The experimental data reflects the best suitable combination of VPA + ZEB is 5 + 0.3mM which has 0.1 CI value. The combination of lower concentration shows complete synergy however, 5 + 0.3mM combination has shown better effects because in cancer therapy, synergism at high dose is more relevant to the therapy than the CI values at low dose.

### 3.2. VPA treatment of colon cancer cell lines down-regulates both KLF4 and CTNNB1 gene expression

After VPA treatment, the expression levels of KLF4 was evaluated using RT-PCR, relative to untreated cells. At both 1.5 mM and 5 mM VPA, KLF4 expression was down regulated in the two colon cancer cell lines, relative to the untreated cells (Fig 1D). However, at 5mM VPA KLF4 was slightly up-regulated (less than one-fold) relative to the lower dose treatment. In the MCF-7 cells, although not significant, treatment with either 1.5 mM or 5 mM VPA up-regulated KLF4 gene expression, with a five-fold difference in the up-regulation of KLF4 at 5 mM when compared to the lower dose (Fig 1D).

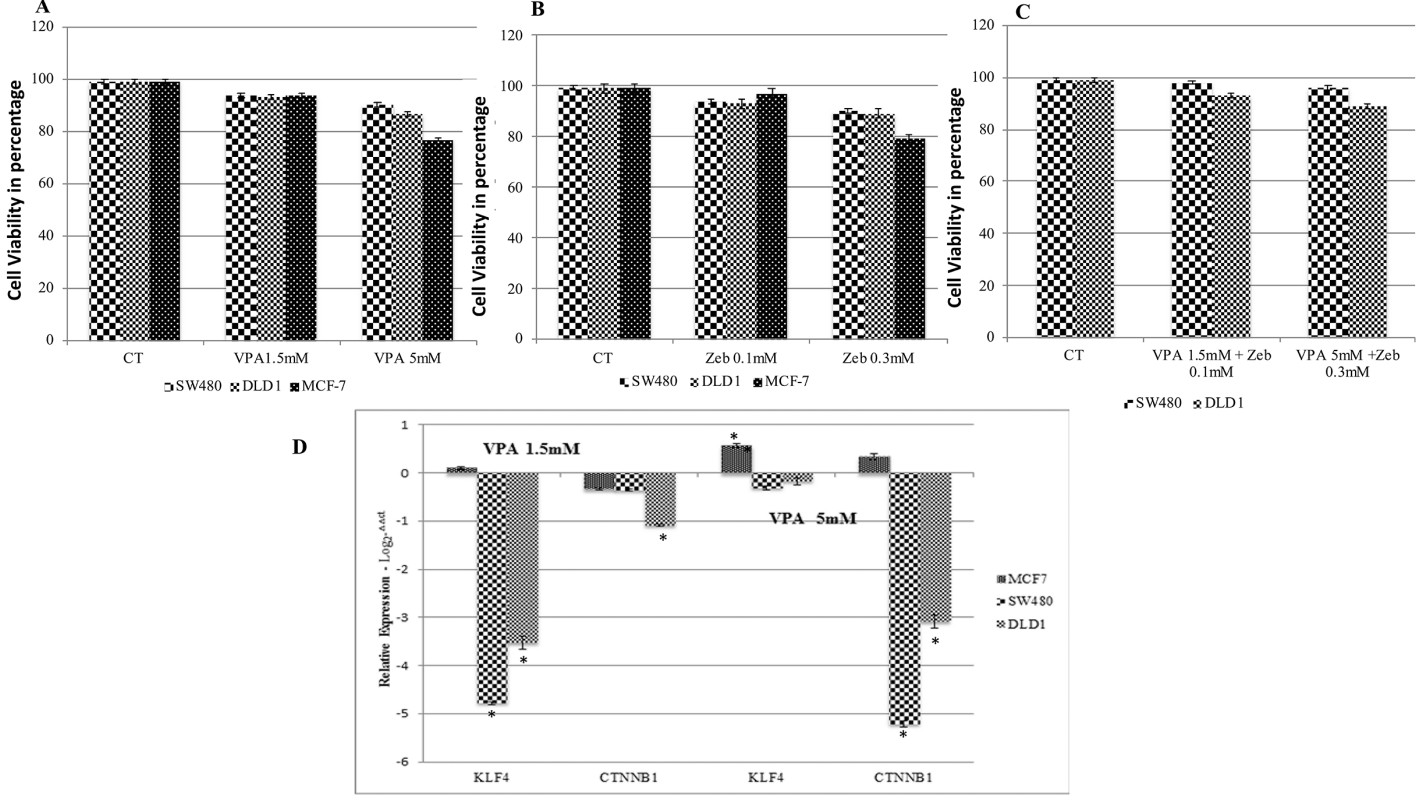

**Fig 1. VPA and ZEB reduce cell viability in the SW480 and DLD-1 colon cancer cells and in the MCF-7 breast cancer cell line. A)** The cells were treated with VPA and **B)** ZEB individually or C) in combination, with the indicated dose of combination for 24 hrs. Carriwer control, DMSO was used a control (CT). **D)** Effect of VPA on KLF4 and CTNNB1 gene expression; Logarithmic expression levels (Y axis) of KLF4 and CTNNB1 in the three cell lines, as determined by quantitative real-time PCR normalized against β-actin (ACTB) (Supplementary data, SD1). Datapoints and error bars represent mean ± **S.**E.M (n = 6). * p < 0.05 relative to the respective DMSO control. ns, statistically non-significant.

Similarly, with regards to CTNNB1 gene expression, both concentrations of VPA down-regulated CTNNB1 gene expression in the SW480 and DLD-1 colon cancer cells (Fig 1D). The transcriptional response was dose-dependent, with a fourteen-fold reduction in CTNNB1 expression in the SW480 cells, and a two-fold decrease in the DLD-1 cells, at 5mM VPA. This transcriptional response is opposite to that of KLF4 expression after VPA treatment. In MCF-7 cells 1.5mM VPA had opposing effects on CTNNB1 expression, while down-regulating this at 1.5mM, but up-regulating expression two-fold at 5 mM.

### 3.3. KLF4 and CTNNB1 protein expression analysis in colon cancer cell lines

Fluorescence microscopy was employed to validate the cellular localization of KLF4 and CTNNB1 before and after the drug treatments. The microscopic visualization of the cells treated with and without drugs clearly showed morphological differences (Fig 2 and Fig.S1 in S1 File), transitioning from a fibroblast-like shape to oval cellular shape and showed a reduced ratio of cytoplasmic to nuclear content. This was evident in the cells treated with lower dose. These changes may relate to an onset of programmed cell death [33].

In the SW480 cell line, although KLF4 was localized to both the cytoplasm and nucleus, localization was more intense in the nucleus. After treatment with 5 mM VPA (Fig 2B) the nuclei were more intensely stained for KLF4, as compared to 1.5mM VPA treated condition (Fig 2A) and the DMSO treated control cells (Fig. S1 in S1 File). In the DLD-1 cells, at 1.5mM VPA, KLF4 expression was strongly expressed around the nuclei with some nuclear localization. At 5mM VPA,

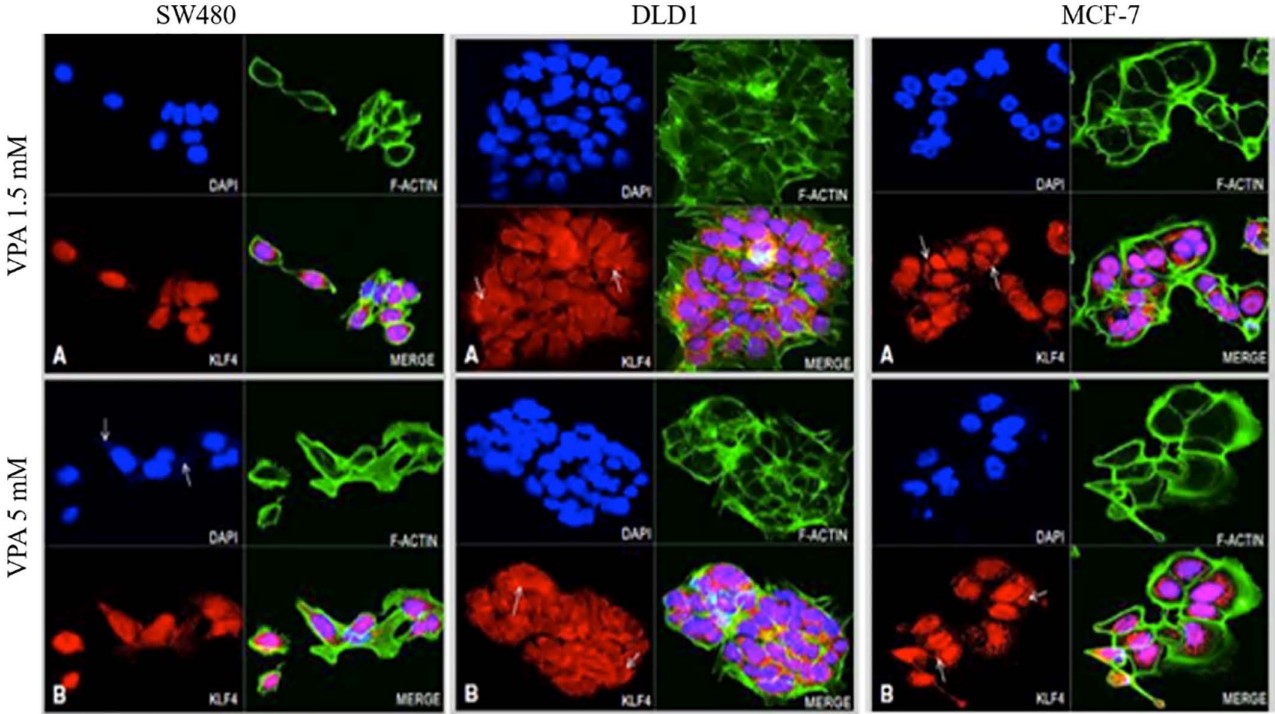

**Fig 2. Effect of VPA on KLF expression in colon cancer cell lines (SW480 and DLD-1) and Breast cancer cell line (MCF-7). A)** Microscope image represents cells treated with 1.5mM VPA and B) represents the 5mM VPA treatment. Arrows in images indicates the KLF4 protein expression and localization. Blue fluorescence represents DAPI stained nuclei; green fluorescence represents the actin filament; red fluorescence represents the KLF protein; pink fluorescence represents the co-staining of KLF4 within the nucleus. MCF-7 was used as a positive control (Original mag. 63X).

cytoplasmic KLF4 appeared reduced with more nuclear staining (pink nuclear signal) occurring. In the MCF-7 cells KLF4 was equally localized to both the cytoplasm and the nuclei at both drug concentrations.

In SW480 cells, β-catenin was specifically localized around the nucleus at 1.5 mM VPA; with a more dispersed cytoplasmic staining at 5 mM VPA. In DLD-1 cells there observed the reduction in cell volume than that of the DMSO treated control cells (Fig.S2 in S1 File), due to a decreased cytoplasmic content. The nucleus was seen to be larger compared to the controls. β-catenin is localized in the nuclear periphery (refer to arrowed Fig. 3A and 3B) and exhibits increased intensity following treatment with 5 mM VPA. MCF-7 cells exhibit an oval to irregular morphology, characterized by a substantial cytoplasmic volume relative to colon cancer cells. There is strong intranuclear and cytoplasmic staining, with particularly intense staining of the nuclear periphery. This staining pattern appears slightly reduced at 5 mM VPA.

### 3.4. ZEB treatment down-regulates KLF4 and CTNNB1 gene expression in colon cancer cell lines

After Zebularine treatment, the expression levels of KLF4 and CTNNB1 were determined using RT-PCR, relative to untreated cells. Zebularine at 0.1 mM down-regulated KLF4 gene expression in both SW480 and DLD-1 cells; while CTNNB1 gene expression was down and up-regulated in SW480 and DLD-1 cells, respectively (Fig 4). At a concentration of 0.3 mM, KLF4 expression exhibited less than one-fold and six-fold up-regulation (p<0.0005) in SW480 and DLD-1 cells, respectively, relative to the low dose treatment. CTNNB1 transcriptional levels were decreased two-fold in the SW480 and DLD-1 cells, compared to the lower dose treatment. This tendency was similar to VPA treatments and contradicts KLF4 gene expression data. In comparison, in the MCF-7, both concentrations up-regulated KLF4 and CTNNB1 expression.

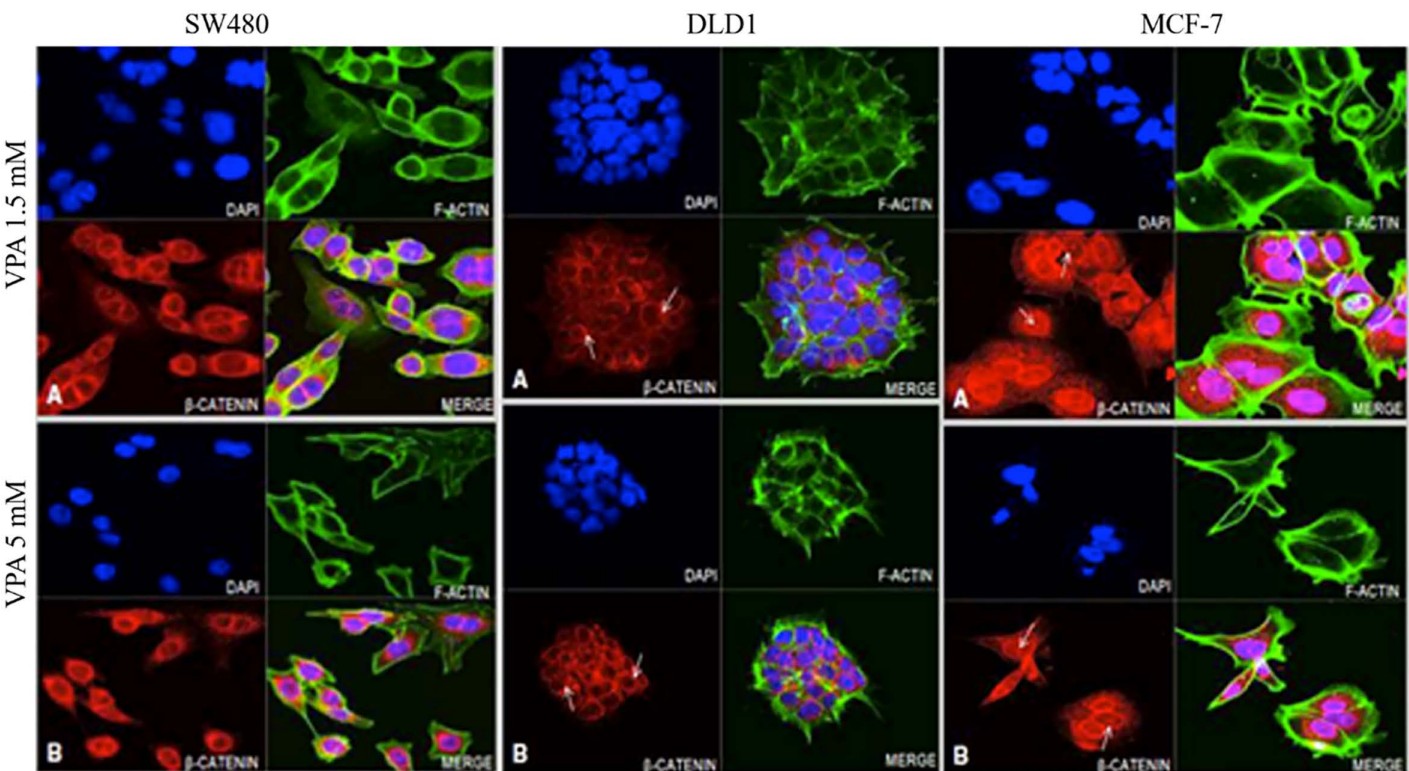

**Fig 3. Effect of VPA on CTNNB1 protein expression in SW480, DLD-1 and MCF-7 cell lines. A)** Microscope image represents cells treated with VPA 1.5 mM and B) represents VPA 5 mM. Arrows in images indicates the CTNNB1 protein expression and localization. Blue fluorescence represents nuclear staining using DAPI; green fluorescence represents the actin filament; Red fluorescence represents the CTNNB1 protein; pink fluorescence represents the co-localization. MCF-7 was used as a positive control (Original mag. 63X).

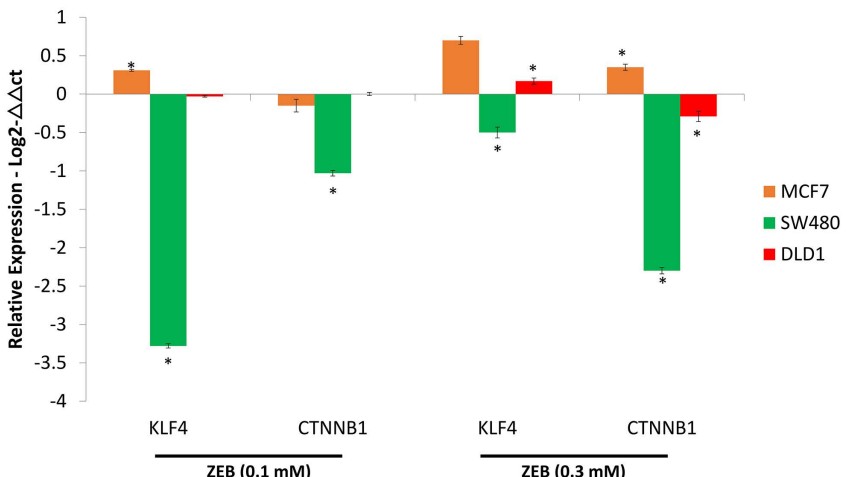

**Fig 4. Effect of Zebularine (ZEB) on KLF4 and CTNNB1 gene expression.** Logarithmic expression levels (Y axis) of KLF4 and CTNNB1 in each cell line using quantitative real-time PCR, normalized against β-actin (ACTB) (Supplementary data, SD2).. Datapoints and error bars represent mean ± **S**.E.M (n = 6). * p < 0.05 relative to the respective DMSO control. ns, statistically non-significant.

## 3.5. Upregulated KLF4 protein expression confirms the DNA demethylation by ZEB

The microscopy images analysis shows the KLF4 protein expression modifications in ZEB treated and untreated cells. The overall irregular morphology is maintained with two distinct concentration treatments, exhibiting the least ratio of cytoplasmic to nuclear content which might be potentially linked to apoptosis. Although KLF4 protein primarily localised and aggregated in the nucleus as noticed in the untreated sample, the intensity of the KLF4 localisation was significantly increased following the treatment with both lower and higher concentrations of ZEB. In SW480 cells, the KLF4 protein exhibits aggregation at the nuclear periphery while maintaining an even distribution throughout the cytoplasm. (Fig 5. I. A and 5.I.B). In DLD-1 cells, the protein is abundant in the nuclear and peripheral regions and evident in Golgi bodies as a C-shaped structure linked to the nucleus (shown in Fig 5.I. A and 5.I.B). These findings from the gene expression data indicates an upregulation in the KLF4 expression than the treatment with lower dose. MCF-7 cells exhibit alterations in morphology from irregular to oval, accompanied by a reduced cell volume than the DMSO treated control (Fig.S3 in S1 File). The high dosage ZEB treatment clearly demonstrated a limited cytoplasmic to nuclear ratio and the presence of membrane blebbing (refer to arrowed Fig 5B), indicating apoptosis. KLF4 was found around the periphery of the nucleus, with the lowest levels spread out in the cytoplasm. The microscopic images showed enlarged nuclei, possibly due to a modification in the status of the chromatin after ZEB treatment, resulting in DNA demethylation and elevated KLF4 transcription levels.

In SW480 cells, the localization of CTNNB1 (β-catenin) closely resembles that of the untreated controls, exhibiting cytoplasmic localization with increased intensity near the cell membrane and nuclear periphery (arrowed Fig 5.II. A and 5.II.B; SW480). Fig 5 reveals an overall irregular cellular morphology in DLD-1; yet the cell volume appears decreased in comparison to the untreated controls (Figs 5.II.A and 5.II.B (DLD-1)). β-catenin protein was found to be predominantly localised in the nuclear periphery (shown by arrows in Fig.5.II.A and 5.II.B (DLD-1)) and minimally distributed in the cytoplasm. There observed no detectable indication of the presence of β-catenin inside the nuclei. Cells exhibits increased compactness with oval shape following high dose treatment (Fig 5.II.B) with ZEB in contrast to low dose treatment (Fig 5.II.A). In MCF-7 cells, β-catenin protein revealed pronounced level of peri-nuclear staining (shown by arrows in Fig 5.II.A and 5.II.B) and cytoplasmic localisation, with modest level of nuclear localisation. The peripheral localisation pattern observed may be associated with chromatin relaxation post treatment, facilitating the peripheral transcription.

## 3.6. Synergistic activity of HDAC and DNMT inhibitors

The effects of co-treatment of VPA and ZEB on gene expression of KLF4 and CTNNB1 were determined using real time PCR. This demonstrated that the combination treatments of either SYN1 with 1.5 mM VPA and 0.1 mM ZEB or SYN2 with 5 mM VPA and 0.3 mM ZEB effectively downregulated the gene expression of both KLF4 and CTNNB1 compared to untreated cells. When using 2-ΔΔCt method, the SYN2 combination treatment exhibited one-fold less in the KLF4 expression up-regulation relative to SYN1 combinatorial treatment (P > 0.001). Consequently, it is noteworthy that treatment using both combinations decreased KLF4 expression. In comparison, the SYN2 combinatorial treatment up-regulated KLF4 expression than the SYN1 (Fig 7). Furthermore, SYN2 treatment of DLD-1 cells, resulted in KLF4 expression being up-regulated with a twenty-two-fold increase, relative to SYN 1 treatment (Fig 6).

In the SW480 cell line, treatment with a high dose of SYN2 shown a twice increase in efficacy in down-regulating the expression of CTNNB1 when compared to the SYN1 combination with low dosage. These results are antagonistic to the KLF4 response to treatment represented in Fig 6. In the DLD-1 cells, SYN2 treatment led to nearly a two-fold reduction in CTNNB1 expression relative to SYN1.

In summary, it is shown that in both cell lines, CTNNB1 expression is antagonistic to KLF4 expression (Fig 6) with CTNNB1 expression decreasing, whilst KLF4 expression increased (Fig 6).

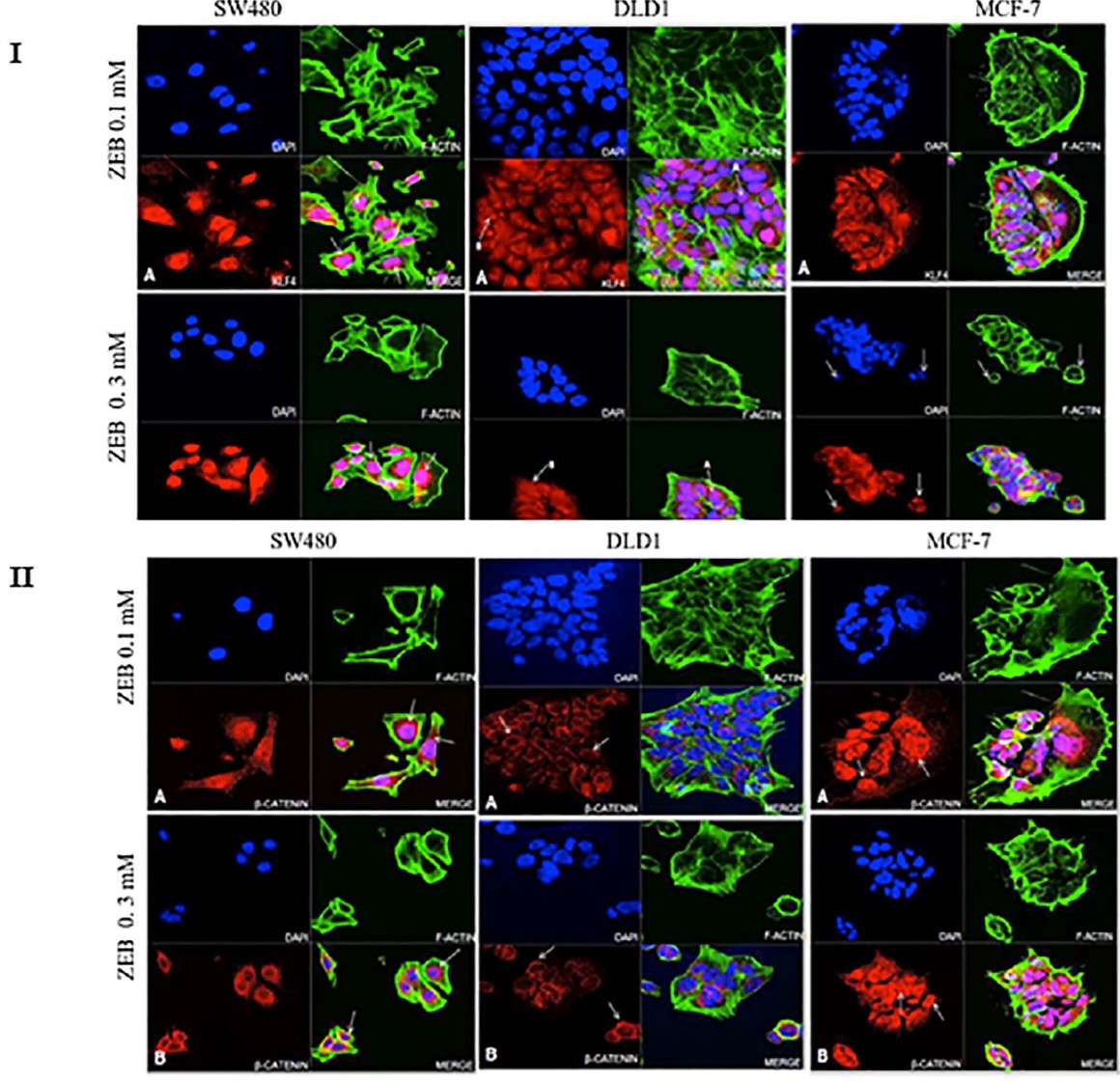

**Fig 5. Confocal microscopy analysis of KLF4 (I) andβ-catenin (II) protein expression in SW480, DLD-1 and MCF-7 cells after ZEB treatment.** In Panel I, Panel A represents cells treated with 0.1mM ZEB; and Panel B represents 0.3mM ZEB. In SW480 cells, arrows indicate nuclear localisation of KLF4, DLD-1- Higher intensity of KLF4 within nucleosomes (arrow A) and Golgi bodies (arrow **B)**. MCF-7- Membrane blebbing was seen in late-stage breast cancer cells (shown in arrow). Blue fluorescence represents nuclear staining using DAPI; green fluorescence represents F-actin filaments; red fluorescence represents the KLF4 protein; pink fluorescence represents the co-localization of KLF4 and DNA (Original mag. 63X). In panel II, Panel A represents cells treated with 0.1mM ZEB; and Panel B represents 0.3mM ZEB. A high intensity of β-catenin staining is seen at nuclear periphery (arrowed). Blue fluorescence represents nuclear staining (DAPI); green fluorescence represents F-actin staining; Red fluorescence represents the β-catenin protein; pink fluorescence represents the co-localization of β-catenin with DNA. MCF-7 was used as a positive control (Original mag. 63X).

### 3.7. Combinatorial effect of VPA and ZEB on KLF protein expression in colon cancer cell lines

In SW480 cells KLF4 was localized to the cytoplasm and the nucleus at both the lower concentration (Fig 7.I.A) and the higher concentration (Fig 7.I.B), with the cells appearing more fibroblast-like at the higher concentration (B). In general, there was a high nuclear concentration of KLF4 with a peri-nuclear accumulation (shown in arrow in Fig 7.I.A and

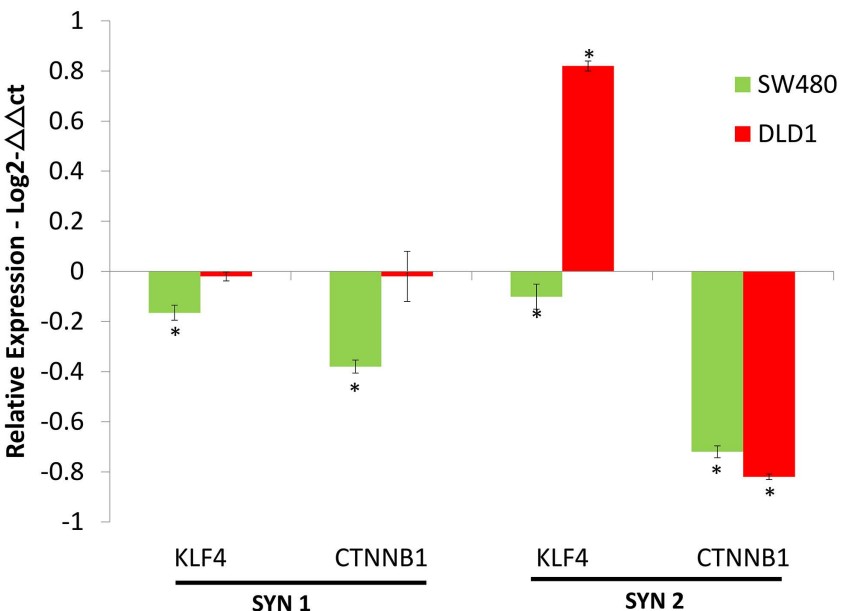

**Fig 6. Expression of KLF4 and CTNNB1 gene expression in colon cancer cells following combination therapy.** Logarithmic expression levels (Y axis) of KLF4 and CTNNB1 in the SW480 and DLD-1 cell lines, normalized against β-actin (ACTB) (Supplementary data, SD3). Datapoints and error bars represent mean±**S**.E.M (n=6). * p<0.05 relative to the respective DMSO control. ns, statistically non-significant.

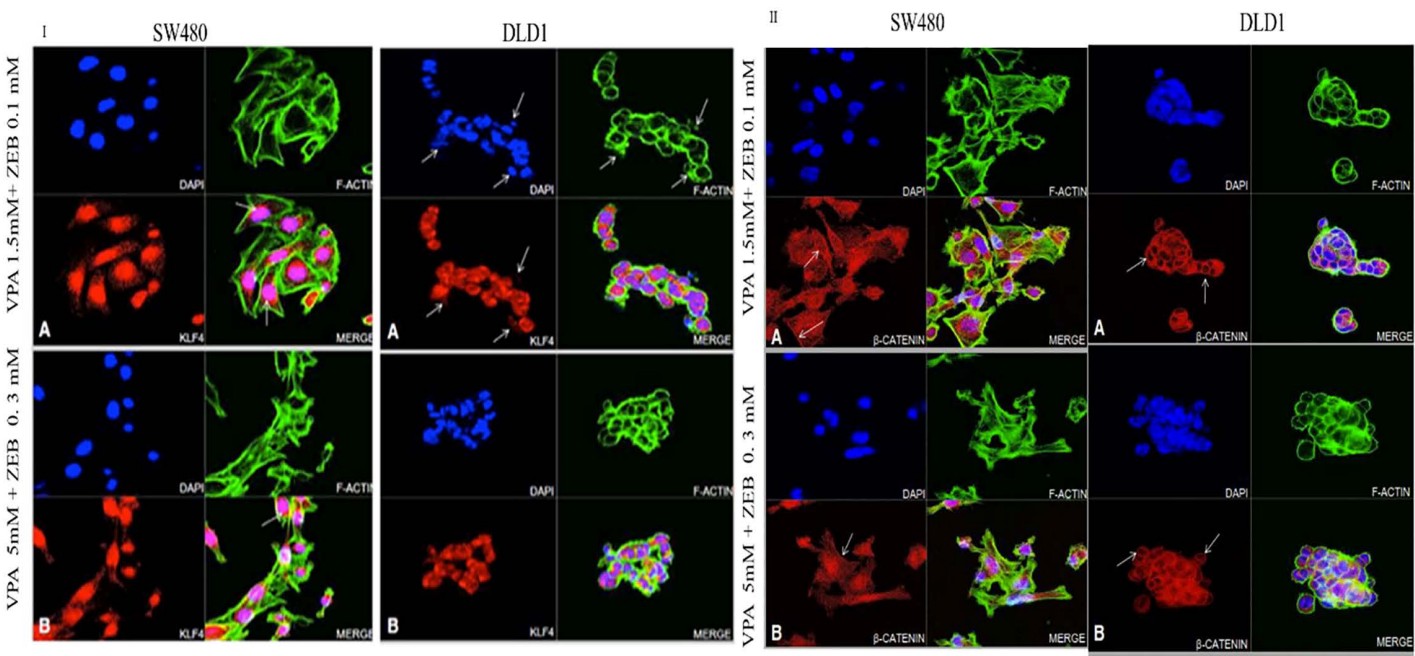

**Fig 7. Synergistic effect of drug treatment on KLF andβ-catenin protein expression in colon cancer cell lines.** 7.I.A) microscope image represents cells treated with 1.5 mM VPA+0.1 mM ZEB; and 7.I.B) 5 mM VPA+0.3 mM ZEB. Blue fluorescence represents DAPI staining of the nuclei; green fluorescence, F actin filament staining; red fluorescence, KLF4 protein; pink fluorescence, represents the co-localization of KLF4 within the nuclei. 7.II.A) represents cells treated with 1.5 mM VPA+0.1 mM ZEB; and 7.II.B) represents 5 mM VPA+0.3 mM ZEB 0.3 mM. Blue fluorescence represents nuclear staining using DAPI; green fluorescence represents F-actin filaments; Red fluorescence represents the β-catenin protein; pink fluorescence represents the co-localization of β-catenin within the nucleus. MCF-7 was used as a positive control (Original mag. 63X).

.I.B). The cells exhibit an oval to round form, and their cell volume significantly reduced following the treatment. KLF4 was localized both to the cytoplasm and to the nuclei, with some reduction in staining at the higher dose (shown in arrow in Fig 7.I.A and Fig 7.I.B).

In SW480 cells, the cytoplasm and nuclei stained intensely for β-catenin with a slight decrease in intensity at the higher treatment concentration (Fig 7.II.A and Fig 7.II.B). In DLD-1 cells there was strong cytoplasmic staining at the lower treatment concentration (Fig 7.II.A), while at the higher concentration, there was increased cytoplasmic and nuclear staining (Fig 7.II.B). These data suggest that combinatorial effect of VPA and ZEB on KLF protein expression was higher that their independent treatment in colon cancer cells.

### 3.8. Single cell quantification of CTNNB1 and KLF4 expression levels

Further analysis of CTNNB1 and KLF4 expression in SW480 and DLD-1 cells in response to the drug treatments was performed using single cell image analysis (Fig 8A). In SW480 cells treated with 5mM VPA and 0.3mM ZEB decreased the expression of CTNNB1 and KLF4; while SYN2 was more effective in KLF4 levels as compared to CTNNB1. A similar pattern was observed with VPA and ZEB treated DLD-1 cells (Fig 8C). However, in comparison to SW480 cells, while SYN-1 and SYN-2 decreased CTNNB1 levels, KLF4 levels were increased. As depicted graphically, (Fig 8B and 8C) Zebularine at 0.1mM seemed to stimulate CTNNB1 levels in both SW480 (Fig 8B) and DLD-1 (Fig 8C) cells. Overall, single cell research demonstrated that the combinatorial treatment impacts SW480 and DLD-1 cells differently, as well as the expression of CTNNB1 and KLF4 proteins in colon cancer cells.

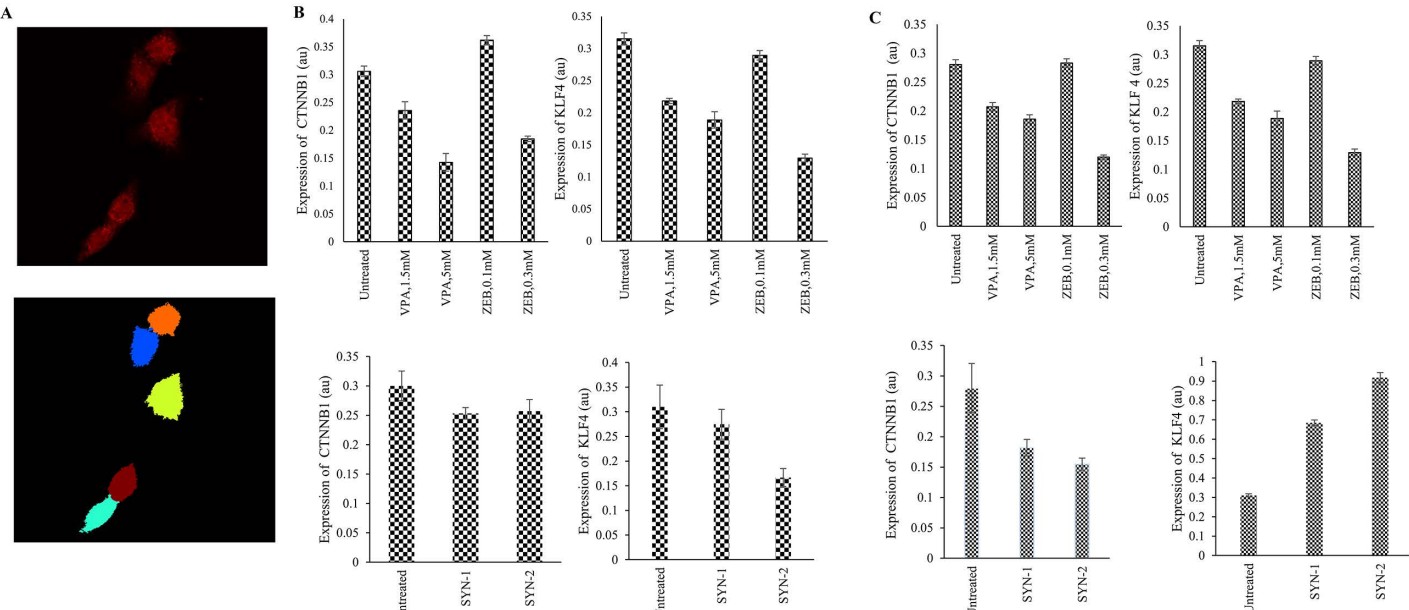

**Fig 8. Single cell protein expression ofβ-catenin and KLF4 in SW480 and DLD-1 cells. A)** The example microscope images of SW480 cells and their segmentation was shown also shown. **B)** The expression of CTNNB1 and KLF4 protein expression in SW480 cells under the pharmacological treatment of VPA, ZEB, SYN-1 and SYN-2. **C)** The expression of CTNNB1 and KLF4 protein expression in DLD-1 cells under the pharmacological treatment of VPA, ZEB, SYN-1 and SYN-2. The example microscope images of DLD-1 cells and their segmentation was shown also shown. The bar graphs represent the mean value of the protein expression (au) with standard error (Y-axis).

## 4. Discussion

Post-translational modification in chromatin structure, such as methylation and histone modifications, plays a vital role in gene regulation. Chromosome alterations may induce tumorigenicity when tumor suppressor genes are rendered inactive under unfavorable epigenetic circumstances [34]. There is a correlation between low levels of histone acetylation and the development of tumors, resulting from elevated HDAC activity, that causes deacetylation [35]. Several HDAC inhibitors are in clinical trials for the treatment of solid and haematological malignancies [36]. Since VPA has a favourable safety profile and a relatively low cost, it has been considered as a candidate drug for various cancer cell types [21,37]. DNA hypermethylation correlates with neoplasia, whereas hypomethylation is observed in benign tissue [38]. Zebularine is a highly stable DNMT preferentially targeting cancer cells, without toxicity in normal cells [39]. VPA induces differentiation and cell cycle arrest by inhibiting HDACs whereas ZEB reactivates silenced genes via the demethylation of tumour suppressors. Each of these anticancer agents has demonstrated efficacy in both in vitro and in vivo studies suggesting that their combined use may amplify their overall therapeutic effects on cancer cells [40,41].

The effects of a DNMTi and an HDACi were evaluated here in colon cancer cell lines, individually and in different dose combinations, with a resulting concentration dependent reduction in cell viability of the colorectal adenocarcinoma cell lines. The combination treatments were more effective in reducing cell viability than either drug alone, as reported in previous studies [42,43]. Associated with this was an alteration in cell shape that may potentially signal an onset of apoptotic induction. DNMTi and HDACi could inhibit tumor cells proliferation by apoptosis in both colon and breast cancer [21,27].

Gene expression profiles data from the current investigation, indicated that both VPA and ZEB exhibited concentration-dependent modulation of gene expression in early-stage and late-stage CRC cells. KLF4 exhibits tumor suppressor characteristics in colon cancer while possessing oncogenic qualities in breast cancer. Subsequently, following the addition of VPA and ZEB, a distinct overexpression of KLF4 was seen in breast cancer cells. Hence, the results indicated that the treatment with lower dosage exhibited stronger potential causing tumorigenic effect on the cells. Upon treatment with higher dosage, it exhibited an increase in apoptosis which was evidenced by the cell viability analysis. The significant overexpression of KLF4 and CTNNB1 was seen with increased VPA compared to ZEB. Treatment with increased concentration was associated with more malignant phenotype, which was evidenced by the presence of KLF4 localised in nuclear region, reported in [44], and this was represented in the confocal microscopy images.

The protein KLF4 is found to be associated with the inhibition of β-catenin, either directly or indirectly, as previously reported [14]. Furthermore, previous reports [45] indicate that the decrease of KLF4 expression correlates with the advancement of CRC; hence, the upregulation of KLF4 by HDACi considerably reduced the malignancy of CRC [28,46]. Theoretically, methylation is lower in early-stage cancer cells, making ZEB less effective than in advanced-stage CRC cells. ZEB decreased KLF4 expression in early-stage CRC cell after low-dose treatment. Subsequently, the high-dose treatments increased expression levels compared to lower doses, but KLF4 expression remained lower than the untreated control; this may be due to, in advanced-stage cancers, the KLF4 gene is more frequently hypermethylated than in early-stage cancers, leading to reduced expression. Treatments with high doses of ZEB significantly increased the KLF4 expression level (Table 2). However, the single-cell analysis showed that the increase in CTNNB1 at 0.1 mM ZEB may be due to cellular heterogeneity, where different cells respond differently to treatment. In whole-population assays, these varied responses are averaged, which can mask opposing effects seen at the single-cell level. At low doses, ZEB may also trigger transient or adaptive responses that temporarily increase CTNNB1 in some cells before inhibition occurs at higher concentrations.

Furthermore, the inverse expression of KLF4 and β-catenin in CRC is directly correlated with the cell viability data, suggesting that the transcriptional pattern plays a role in apoptosis. This concept has been supported by several research indicating that the suppression of β-catenin in CRC may facilitate apoptosis. The down-regulation of β-catenin, potential regulator of Wnt signaling pathway, may occur due to Wnt pathway blockage, thereby increasing the risk of cell death [47,48].

**Table 2. Summary of mRNA and protein expression changes following VPA and ZEB treatment-** mRNA (qPCR) and protein (immunofluorescence) expression changes of KLF4 and CTNNB1 in early-stage (SW480) and late-stage (DLD-1) colorectal cancer cells following treatment with valproic acid (VPA), zebularine (ZEB), and their combination. Arrows indicate relative upregulation or downregulation compared to untreated controls.

| Cell Line | Treatment | KLF4 mRNA (qPCR) | KLF4 Protein (IF) | CTNNB1 mRNA (qPCR) | CTNNB1 Protein (IF) |
|---|---|---|---|---|---|
| **SW480** (Early-stage) | VPA (low dose) | ↓ | ↔ / slight ↑ | ↓ | ↓ |
| | VPA (high dose) | ↓ | ↑ **(nuclear)** | ↓↓ | ↓ |
| | ZEB (low dose) | ↓ | ↑ | ↓ | ↓ |
| | ZEB (high dose) | ↔ / ↓ | ↑ | ↓↓ | ↓↓ |
| | VPA+ZEB (low dose) | ↔ | ↔ | ↓ | ↓ |
| | VPA+ZEB (high dose) | ↔ / ↓ | ↓ | ↓↓ | ↓↓ |
| **DLD-1** (Late-stage) | VPA (low dose) | ↓ | ↔ | ↓ | ↓ |
| | VPA (high dose) | ↓ | ↑ **(nuclear)** | ↓↓ | ↓ |
| | ZEB (low dose) | ↓ | ↑ | ↓ | ↓ |
| | ZEB (high dose) | ↔ | ↑ | ↓↓ | ↓↓ |
| | VPA+ZEB (low dose) | ↑ | ↔ | ↓ | ↓ |
| | VPA+ZEB (high dose) | ↑↑ | ↔ / modest ↑ | ↓↓ | ↓↓ |

A parallel study conducted on other cancer cell lines treated with Sulindac showed similar results, indicating that β-catenin was instrumental in preventing apoptosis. Significantly, the overexpression of β-catenin is solely a consequence of a hyperactive Wnt pathway [49]. KLF4 is specifically regulated in breast cancer cells, functioning as an oncogene. Consequently, elevated expression of KLF4 in MCF-7 cells is correlated with a aggressive phenotype [28,46]. A low dose of HDACi treatment led to a marginal increase the expression of KLF4 compared to the control. Higher doses of HDACi may cause tumorigenic activity or apoptosis, depending on its interaction with β-catenin.

The KLF4 mRNA downregulation and the increased nuclear KLF4 protein levels following VPA or ZEB treatment can be explained by post-transcriptional and post-translational regulatory mechanisms that are well documented in the literature. KLF4 is known to be tightly regulated at multiple levels, including mRNA stability, translational control, protein stabilization, and nuclear retention. HDAC inhibitors such as VPA can enhance protein acetylation, leading to increased protein stability and nuclear accumulation without a corresponding increase in mRNA levels. Similarly, DNA methylation inhibitors like ZEB may alter the expression of microRNAs or translational regulators that suppress KLF4 translation, resulting in enhanced protein expression despite reduced transcript levels. Previous studies have reported poor correlation between KLF4 mRNA and protein abundance due to ubiquitin-proteasome–mediated degradation, altered half-life, and context-dependent nuclear localization of KLF4 protein [50–52].

The differential effects observed under combinatorial VPA+ZEB treatment likely reflect cell line–specific regulatory mechanisms. In SW480 cells, high-dose combination treatment may enhance post-translational degradation or impair nuclear retention of KLF4 protein, leading to reduced protein intensity despite transcriptional activity. In contrast, DLD-1 cells may exhibit transcriptional de-repression of KLF4 due to cumulative epigenetic effects, highlighting stage- and context-dependent regulation of KLF4 in colorectal cancer.

The current work demonstrated a considerable upregulation of KLF4 following high doses in late-stage CRC cells treated synergistically with VPA and ZEB. The demethylating effect of the combined treatment appears to be the main factor influencing KLF4 expression. ZEB and VPA may also influence the Wnt pathway by altering key regulatory components, including activation of the negative Wnt regulator GSK-3 and WNT9A [53]. This could trigger a cascade of downstream effects, ultimately leading to downregulation of β-catenin and reduced cell proliferation, thereby promoting apoptotic processes [54]. Overall, the combination of VPA and ZEB significantly increased KLF4 expression compared

to individual treatments, particularly at higher concentrations, further supporting their role as epigenetic modulators in colorectal cancer (Fig 9).

## Conclusion

Our data describe the effects of valproic acid (VPA) and zebularine (ZEB), tested alone and in combination, on the transcription of KLF4 and β-catenin (CTNNB1) in colorectal cancer cell lines SW480 and DLD-1, as well as in control breast cancer cells, MCF-7. Expression of these transcription factors was analyzed by real-time RT-PCR and at the subcellular level using confocal microscopy. Additionally, the effects of VPA and ZEB on cell viability were assessed. We observed that high concentrations of VPA and ZEB increased KLF4 mRNA levels by approximately 2-fold compared with lower concentrations in early-stage SW480 cells, while the combination of the two agents at high concentrations further elevated KLF4 expression relative to the lower-dose combination. CTNNB1 was downregulated relative to KLF4. In summary, our findings suggest that the combinatorial repurposing of VPA and ZEB may serve as suboptimal anti-cancer agents, highlighting their role as epigenetic modulators in colorectal cancer.

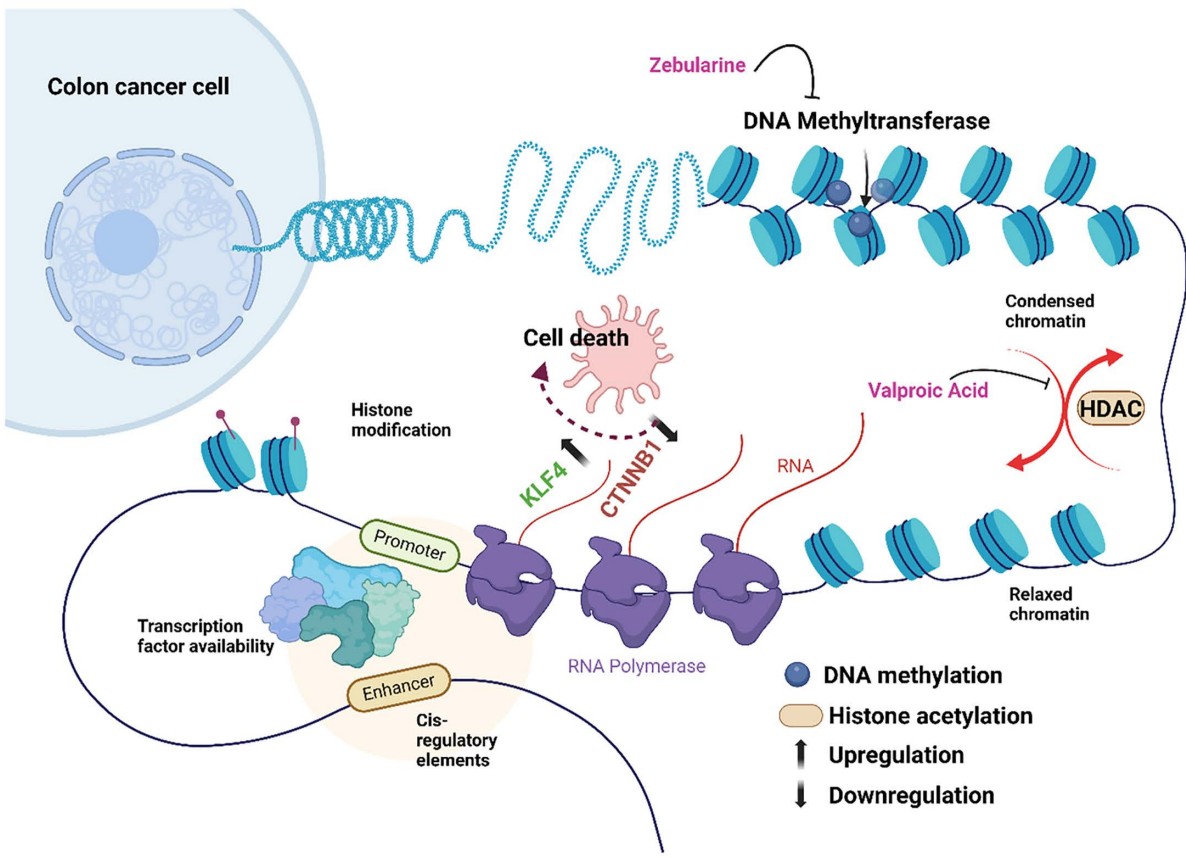

**Fig 9. The schematic representation of the biological mechanism of histone deacetylase inhibitor (HDACi), Valproic acid (VPA) and the DNA methyltransferase inhibitor, Zebularine (ZEB) on the expression of the tumour suppressor gene Krüppel-like factor 4 (KLF4) and the oncogene, CTNNB1 (β-catenin) in colon cancer cells.**

## Supporting Information

**S1 File. Supplementary File.**
(DOCX)

## Author contributions

**Conceptualization:** Clement Penny.

**Data curation:** Jeyalakshmi Kandhavelu, Kumar Subramanian, Natanya Moodley.

**Formal analysis:** Jeyalakshmi Kandhavelu, Kumar Subramanian, Natanya Moodley.

**Funding acquisition:** Meenakshisundaram Kandhavelu, Paul Ruff, Clement Penny.

**Investigation:** Jeyalakshmi Kandhavelu, Meenakshisundaram Kandhavelu, Paul Ruff, Clement Penny.

**Methodology:** Jeyalakshmi Kandhavelu, Kumar Subramanian, Natanya Moodley.

**Project administration:** Meenakshisundaram Kandhavelu, Paul Ruff, Clement Penny.

**Resources:** Kasim S. Abass.

**Software:** Kasim S. Abass, Chandrabose Sureka.

**Supervision:** Meenakshisundaram Kandhavelu, Paul Ruff, Clement Penny.

**Validation:** Jeyalakshmi Kandhavelu, Kumar Subramanian.

**Visualization:** Jeyalakshmi Kandhavelu, Kumar Subramanian, Natanya Moodley, Kasim S. Abass, Chandrabose Sureka.

**Writing – original draft:** Jeyalakshmi Kandhavelu, Kumar Subramanian, Chandrabose Sureka, Meenakshisundaram Kandhavelu, Paul Ruff, Clement Penny.

**Writing – review & editing:** Jeyalakshmi Kandhavelu, Kumar Subramanian, Kasim S. Abass, Chandrabose Sureka, Meenakshisundaram Kandhavelu, Paul Ruff, Clement Penny.

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
