## [Decision Letter · Decision Letter 0]

11 Sep 2025

PONE-D-25-45571

Combinatorial drug repurposing of Valproic acid and Zebularine regulates Krüppel-like factor 4 and β-catenin expression in colon cancer cells

PLOS ONE

Dear Dr. Kandhavelu,

Thank you for submitting your manuscript to PLOS ONE. After careful consideration, we have decided that your manuscript does not meet our criteria for publication and must therefore be rejected.

Specifically, the reviewers and editors found that the manuscript lacks proper controls and statistical analysis to support the conclusions. We are sorry that we cannot be more positive on this occasion, but hope that you appreciate the reasons for this decision.

Kind regards,

Chunming Liu

Academic Editor

PLOS ONE

Reviewers' comments:

Reviewer's Responses to Questions

**Comments to the Author**

1. Is the manuscript technically sound, and do the data support the conclusions?

Reviewer #1: No

2. Has the statistical analysis been performed appropriately and rigorously? 

Reviewer #1: No

3. Have the authors made all data underlying the findings in their manuscript fully available?

Reviewer #1: Yes

4. Is the manuscript presented in an intelligible fashion and written in standard English?

Reviewer #1: No

5. Review Comments to the Author

Reviewer #1: This manuscript reported a combination drug therapy by repurposing Valproic acid and Zebularine to regulate Krüppel-like

factor 4 and β-catenin expression in colon cancer cells. However, it does not meet the publishing criteria of PLOS ONE.

1, Figure 1 shows no synergistic effects between Valproic acid and Zebularine in cell viability even though the authors claimed otherwise.

2, Figures 2,3,5 lack vehicle controls.

3, Figures 1,4,6 are missing statistical analysis.

6. PLOS authors have the option to publish the peer review history of their article (what does this mean?). If published, this will include your full peer review and any attached files.

Reviewer #1: No

- - - - -

---

## [Decision Letter · Decision Letter 1]

10 Dec 2025

PONE-D-25-45571R1Combinatorial drug repurposing of Valproic acid and Zebularine regulates Krüppel-like factor 4 and β-catenin expression in colon cancer cellsPLOS One

Dear Dr. Kandhavelu,

Thank you for submitting your manuscript to PLOS ONE. After careful consideration, we feel that it has merit but does not fully meet PLOS ONE’s publication criteria as it currently stands. Therefore, we invite you to submit a revised version of the manuscript that addresses the points raised during the review process. 

Provide appropriate controls and inclusion of additional assays to attribute the tumour suppressive property of KLF4.

A higher level of validation shall be provided for the observed changes in gene expression.

Epigenetic changes observed shall be justified with additional experimentation using ChIP.

Though novelty is not a prime concern to publish in PLoS ONE, it is essential to provide sufficient evidence on the claims made so that the study will contribute to the further advancement of the knowledge in this domain.

Inconsistency in the expression of KLF4 with the drug dosage should be thoroughly addressed.

Efforts have to be made to provide mechanistic insights to support the claims made.

We look forward to receiving your revised manuscript.

Kind regards,

Suresh Yenugu

Academic Editor

PLOS One

When submitting your revision, we need you to address these

additional requirements.

1. Please ensure that your manuscript meets PLOS ONE's style

requirements, including those for file naming. The PLOS ONE style templates can

be found at

and

“We would like to

acknowledge Natanya Moodley for manual support during the execution of research

methodology. The study was supported by funding from the Cancer Association of

South Africa (CANSA); and the Medical Research Council (MRC) of South Africa to

the Wits/MRC Common Epithelial Cancer Research Centre. JK is grateful to the

National Research Foundation (NRF) of South Africa, to the Griffin Trust Fund,

University of the Witwatersrand and the Faculty of Health Sciences Research

Committee (FRC) for funding. We also would like to thank Department of Internal

Medicine, University of the Witwatersrand and all the funding agency.”

Please provide an amended statement that declares all the

funding or sources of support (whether external or internal to your

organization) received during this study, as detailed online in our guide for

authors at http://journals.plos.org/plosone/s/submit-now. Please also include the statement “There was

no additional external funding received for this study.” in your updated

Funding Statement.

Please include your amended Funding Statement within your

cover letter. We will change the online submission form on your behalf.

“We would like to

acknowledge Natanya Moodley for manual support during the execution of research

methodology. The study was supported by funding from the Cancer Association of

South Africa (CANSA); and the Medical Research Council (MRC) of South Africa to

the Wits/MRC Common Epithelial Cancer Research Centre. JK is grateful to the

National Research Foundation (NRF) of South Africa, to the Griffin Trust Fund,

University of the Witwatersrand and the Faculty of Health Sciences Research

Committee (FRC) for funding. We also would like to thank Department of Internal

Medicine, University of the Witwatersrand and all the funding agency.”

Please state what role the funders took in the study. If the funders had no role, please state:

"The funders had no role in study design, data collection and analysis,

decision to publish, or preparation of the manuscript."

If this statement is not correct you must amend it as

needed.

Please include this amended Role of Funder statement in your

cover letter; we will change the online submission form on your behalf.

4. We note that your Data

Availability Statement is currently as follows: [All

relevant data are within the manuscript and its Supporting Information files.]

Please confirm at this time whether or not your submission

contains all raw data required to replicate the results of your study. Authors

must share the “minimal data set” for their submission. PLOS defines the

minimal data set to consist of the data required to replicate all study

findings reported in the article, as well as related metadata and methods

(https://journals.plos.org/plosone/s/data-availability#loc-minimal-data-set-definition).

The values behind the means, standard deviations and other

measures reported;

The values used to build graphs;

The points extracted from images for analysis.

Authors do not need to submit their entire data set if only

a portion of the data was used in the reported study.

If your submission does not contain these data, please

either upload them as Supporting Information files or deposit them to a stable,

public repository and provide us with the relevant URLs, DOIs, or accession

numbers. For a list of recommended repositories, please see

https://journals.plos.org/plosone/s/recommended-repositories.

If there are ethical or legal restrictions on sharing a

de-identified data set, please explain them in detail (e.g., data contain

potentially sensitive information, data are owned by a third-party

organization, etc.) and who has imposed them (e.g., an ethics committee).

Please also provide contact information for a data access committee, ethics

committee, or other institutional body to which data requests may be sent. If

data are owned by a third party, please indicate how others may request data

access.

5. Thank you for stating the following in your manuscript:

“We

would like to acknowledge Natanya Moodley for manual support during the

execution of research methodology. The study was supported by funding from the

Cancer Association of South Africa (CANSA); and the Medical Research Council

(MRC) of South Africa to the Wits/MRC Common Epithelial Cancer Research Centre.

JK is grateful to the National Research Foundation (NRF) of South Africa, to

the Griffin Trust Fund, University of the Witwatersrand and the Faculty of

Health Sciences Research Committee (FRC) for funding. We also would like to

thank Department of Internal Medicine, University of the Witwatersrand and all

the funding agency. “

We note that you have provided additional information within

the Acknowledgements Section that is not currently declared in your Funding

Statement. Please note that funding information should not appear in the

Acknowledgments section or other areas of your manuscript. We will only publish

funding information present in the Funding Statement section of the online

submission form.

Please remove any funding-related text from the manuscript

and let us know how you would like to update your Funding Statement. Currently,

your Funding Statement reads as follows:

“We would like to acknowledge Natanya Moodley for manual

support during the execution of research methodology. The study was supported

by funding from the Cancer Association of South Africa (CANSA); and the Medical

Research Council (MRC) of South Africa to the Wits/MRC Common Epithelial Cancer

Research Centre. JK is grateful to the National Research Foundation (NRF) of

South Africa, to the Griffin Trust Fund, University of the Witwatersrand and

the Faculty of Health Sciences Research Committee (FRC) for funding. We also

would like to thank Department of Internal Medicine, University of the

Witwatersrand and all the funding agency.”

Please include your amended statements within your cover

letter; we will change the online submission form on your behalf.

6. Please include captions for your Supporting Information

files at the end of your manuscript, and update any in-text citations to match

accordingly. Please see our Supporting Information guidelines for more

information: http://journals.plos.org/plosone/s/supporting-information.

Additional Editor Comments (if provided):

Reviewers' comments:

Reviewer's Responses to Questions

**Comments to the Author**

1. If the authors have adequately addressed your comments raised in a previous round of review and you feel that this manuscript is now acceptable for publication, you may indicate that here to bypass the “Comments to the Author” section, enter your conflict of interest statement in the “Confidential to Editor” section, and submit your "Accept" recommendation.

Reviewer #2: (No Response)

Reviewer #3: All comments have been addressed

2. Is the manuscript technically sound, and do the data support the conclusions?

Reviewer #2: Partly

Reviewer #3: No

3. Has the statistical analysis been performed appropriately and rigorously? 

Reviewer #2: Yes

Reviewer #3: Yes

4. Have the authors made all data underlying the findings in their manuscript fully available?

Reviewer #2: Yes

Reviewer #3: Yes

5. Is the manuscript presented in an intelligible fashion and written in standard English?

Reviewer #2: Yes

Reviewer #3: Yes

6. Review Comments to the Author

Reviewer #2: General Comments to the Authors:

1. The introduction is generally well structured; however, it would benefit from a more comprehensive discussion of the clinical symptoms, underlying causes, and major risk factors of colon cancer. Furthermore, a clearer linkage between these aspects and the rationale, objectives, and significance of the current study should be established to strengthen the contextual foundation of the manuscript.

2. The manuscript should maintain consistent scientific terminology throughout. In particular, β-catenin is referred to interchangeably as CTNNB1 and β-catenin, which may cause confusion and should be standardized.

3. The vehicle control images corresponding to the confocal microscopy experiments should be assembled and presented together in a single figure to improve clarity, organization, and comparative interpretation of the imaging data.

4. Vehicle control data appear to be missing from Figures 1D, 4, and 6. In the absence of these essential controls, the comparison of gene expression levels and treatment effects is scientifically inappropriate and weakens the reliability of the results.

5. In line 109 of the Introduction, the authors state that KLF4-overexpressing MCF-7 cells were used; however, there is no experimental evidence or methodological description confirming KLF4 overexpression. This inconsistency should be clarified and adequately supported.

6. The tumour-suppressive role attributed to KLF4 requires stronger experimental validation. Inclusion of apoptosis-specific assays, such as flow cytometry (FACS) analysis and Western blotting of apoptotic markers, is necessary because immunofluorescence alone is insufficient to conclusively demonstrate apoptotic induction.

7. The reported changes in gene expression should be supported by protein-level validation. Western blotting and relevant transfection-based functional assays are required to confirm that transcriptional changes translate into meaningful biological effects.

8. The reduction in cell proliferation does not appear to be proportionate to the reported increase in KLF4 expression. Given the modest decrease in cell viability relative to control, the claim that these effects are KLF4-dependent (as stated in lines 549–550) appears overstated and requires additional experimental justification. The use of the term “efficient” is also not adequately supported by the magnitude of the observed effects.

9. The proposed epigenetic effects of ZEB and VPA on histone modifications should be validated using chromatin immunoprecipitation (ChIP) assays to directly measure alterations in HDAC-associated histone marks. Reliance solely on qPCR data is insufficient to draw firm conclusions about the efficacy of the VPA+ZEB combination in colorectal cancer.

10. There appears to be a misrepresentation of concentration values in line 384, which should be carefully corrected to ensure the accuracy and reproducibility of the experimental conditions.

11. The study currently lacks direct mechanistic evidence and relies heavily on previously published literature to support its conclusions. For a manuscript that claims novelty, original experimental validation of the proposed mechanisms is required rather than extrapolation from prior reports.

Overall, while the study addresses an interesting and potentially relevant research question, the manuscript requires substantial revision. Critical methodological clarifications, inclusion of appropriate controls, and stronger mechanistic and protein-level validation are essential before the conclusions can be considered reliable.

Reviewer #3: There are several critical inconsistencies between the mRNA data, protein-level observations, dosing rationale, and the mechanistic interpretation. These issues collectively undermine the reliability of the study’s conclusions.

1. Multiple discrepancies remain between qPCR data and confocal imaging results.

• KLF4 mRNA is downregulated by VPA in both SW480 and DLD-1 cells (Fig. 1D), yet nuclear KLF4 protein staining increases markedly at 5 mM VPA (Fig. 2B).

• Low-dose ZEB decreases KLF4 mRNA (Fig. 4, pages 15–16), but KLF4 protein intensity increases at both low and high ZEB doses (Fig. 5.I.A and 5.I.B, pages 17–18).

• Under combinatorial treatment, SW480 cells show reduced KLF4 protein intensity at high dose (Fig. 7.I.B), while DLD-1 cells show substantial KLF4 upregulation at the mRNA level (Fig. 6, page 20).

These contradictions are neither addressed nor explained. No mechanistic rationale, methodological clarification, or acknowledgment of potential post-transcriptional or post-translational regulation is provided. This creates fundamental uncertainty about the validity of the central conclusions.

2. Although the manuscript lists the doses used (1.5 mM / 5 mM VPA and 0.1 mM / 0.3 mM ZEB), the authors provide no scientific justification for selecting these concentrations.

• No EC50/IC50 values are referenced.

• No preliminary optimization or toxicity data are presented.

• No supporting literature is cited.

Without proper justification, it is unclear whether the chosen concentrations reflect physiologic relevance, mechanistic intent, or simply reflect cytotoxicity-driven effects. This remains a major methodological limitation.

3. The manuscript continues to assert mechanistic interactions that are not experimentally demonstrated:

• Fig. 9 (page 27) is entirely conceptual and is not supported by any functional data generated in the study.

• The Discussion (pages 23–26) repeatedly presents causal interpretations, particularly regarding Wnt signaling, apoptosis induction, and epigenetic regulation, that are not proven experimentally.

• The proposed antagonistic relationship between KLF4 and CTNNB1 is not consistently supported by the gene expression data in Fig. 6 or the single-cell protein data in Fig. 8.

• Crucially, no functional assays (e.g., KLF4 knockdown, β-catenin reporter assays, rescue experiments, chromatin studies) were performed to validate the proposed mechanism.

7. PLOS authors have the option to publish the peer review history of their article (what does this mean?). If published, this will include your full peer review and any attached files.

Reviewer #2: No

Reviewer #3: **Yes:** Abhijit Chakraborty

---

## [Author Response · Author response to Decision Letter 2]

19 Feb 2026

Thanks for the reviewer comments. The response to reviewer comments is attached as seperate file.

---

## [Decision Letter · Decision Letter 2]

15 Mar 2026

PONE-D-25-45571R2Combinatorial drug repurposing of Valproic acid and Zebularine regulates Krüppel-like factor 4 and β-catenin expression in colon cancer cellsPLOS One

Dear Dr. Kandhavelu,

Thank you for submitting your manuscript to PLOS ONE. After careful consideration, we feel that it has merit but does not fully meet PLOS ONE’s publication criteria as it currently stands. Therefore, we invite you to submit a revised version of the manuscript that addresses the points raised during the review process.

We look forward to receiving your revised manuscript.

Kind regards,

Suresh Yenugu

Academic Editor

PLOS One

**Journal Requirements:**

**Additional Editor Comments:**

Note that the suggestions of the reviewers have to be taken in all seriousness and modifications made, especially when claims are made on the mechanistic action in the absence of substantial data. Changes to the introduction, rationale and discussion should be made by citing appropriate references.

Reviewers' comments:

Reviewer's Responses to Questions

**Comments to the Author**

1. If the authors have adequately addressed your comments raised in a previous round of review and you feel that this manuscript is now acceptable for publication, you may indicate that here to bypass the “Comments to the Author” section, enter your conflict of interest statement in the “Confidential to Editor” section, and submit your "Accept" recommendation.

Reviewer #2: (No Response)

Reviewer #3: All comments have been addressed

2. Is the manuscript technically sound, and do the data support the conclusions?

Reviewer #2: Partly

Reviewer #3: Partly

3. Has the statistical analysis been performed appropriately and rigorously? 

Reviewer #2: Yes

Reviewer #3: I Don't Know

4. Have the authors made all data underlying the findings in their manuscript fully available?

Reviewer #2: Yes

Reviewer #3: Yes

5. Is the manuscript presented in an intelligible fashion and written in standard English?

Reviewer #2: Yes

Reviewer #3: No

6. Review Comments to the Author

Reviewer #2: The revise manuscript does not still fully address the particular major comments mentioned in previous review which plays a significant role in proving the credibility of the study.

Reviewer #3: The authors have made some effort to address my previous concerns, but several critical issues remain unresolved or insufficiently addressed. My overall assessment is that the manuscript still requires major revision before it can be considered for publication.

Major Comments:

1. The authors added a paragraph to the Discussion invoking post-transcriptional and post-translational regulatory mechanisms to explain the discordances between KLF4 mRNA and protein levels. While this is a plausible general explanation, it remains entirely speculative and unsupported by any data generated in this study. No mRNA stability assays, polysome profiling, proteasome inhibition experiments, or microRNA data are presented. The response cites references 49-51 for these mechanisms, but these references do not appear in the manuscript's reference list this is a significant error that must be corrected. Furthermore, for the combinatorial treatment discordance (SW480 vs. DLD-1), the authors offer only a qualitative, cell-line-specific rationale without experimental substantiation. The discordances between mRNA and protein data remain a fundamental unresolved issue that meaningfully weakens the study's conclusions.

2. The authors have added a paragraph to the Introduction justifying their concentration selections. This is an improvement. However, the justification cites references [24–28] that are not present in the reference list as numbered the numbering appears shifted and inconsistent with the manuscript body. This must be corrected. Additionally, the authors do not address whether VPA at 1.5–5 mM is clinically achievable in plasma or tissue — this is important context for a drug repurposing claim.

3. The authors replaced the schematic diagram (Fig. 9) with Table 2. While this is a positive step toward transparency, the core issue remains: causal mechanistic claims about Wnt signaling, apoptosis induction, and the KLF4-β-catenin antagonism continue to be made in the Discussion without any supporting functional data. No knockdown experiments, reporter assays, rescue experiments, or ChIP data have been added. The removal of the schematic is appreciated, but it does not resolve the problem of unsupported mechanistic language. The authors should clearly reframe the Discussion using appropriately hedged, correlative language, and distinguish firmly between what was observed versus what is speculated.

Minor Comments:

1. The text still refers to "SYN 2 with 25 mM VPA" — this appears to be a persistent error (should be 5 mM VPA). The authors acknowledged this in their response but it must be verified as corrected in the final manuscript.

2. Line 359 refers to "HT29 and DLD-1 cells" when the study uses SW480 and DLD-1. This error remains uncorrected in the R2 manuscript and represents a serious inconsistency.

3. The abstract (line 36 of the R2 manuscript) still refers to "about one-fold" increase in DLD-1 cells in one place, while the body consistently reports a 22-fold increase. This needs reconciliation.

4. The single-cell quantification data in Figure 8 show that ZEB at 0.1 mM appears to stimulate CTNNB1 levels in both SW480 and DLD-1 cells, which is contradictory to the proposed anti-cancer mechanism. This is acknowledged briefly in the Results but not adequately discussed.

5. The Dunnett's test is appropriate for the comparisons described, but the manuscript continues to report some p-values inconsistently (e.g., "p > 0.0005" in section 3.4, which appears to be a typographical error and should likely be p < 0.0005).

6. The manuscript still contains grammatical errors and unclear phrasing throughout that impede scientific comprehension (e.g., "huge increase to about one-fold," "confrontational interaction," "injection of VPA and ZEB"). Professional English editing is still needed.

The authors have made partial progress, but the manuscript continues to make mechanistic claims unsupported by functional data, contains unresolved discordances between mRNA and protein results, has uncorrected errors (including a cell line identification error in Section 3.8 and missing/incorrect references), and retains language issues. The study's conceptual contribution exploring VPA and ZEB combination effects on KLF4 and β-catenin in CRC remains of modest interest, but the conclusions must be more carefully aligned with the actual data, and the identified errors corrected before this work can be published.

7. PLOS authors have the option to publish the peer review history of their article (what does this mean?). If published, this will include your full peer review and any attached files.

Reviewer #2: No

Reviewer #3: **Yes:** Abhijit Chakraborty

---

## [Decision Letter · Decision Letter 3]

14 Apr 2026

Combinatorial drug repurposing of Valproic acid and Zebularine regulates Krüppel-like factor 4 and β-catenin expression in colon cancer cells

PONE-D-25-45571R3

Dear Dr. Kandhavelu,

We’re pleased to inform you that your manuscript has been judged scientifically suitable for publication and will be formally accepted for publication once it meets all outstanding technical requirements.

Kind regards,

Suresh Yenugu

Academic Editor

PLOS One

Additional Editor Comments (optional):

Reviewers' comments:

Reviewer's Responses to Questions

**Comments to the Author**

1. If the authors have adequately addressed your comments raised in a previous round of review and you feel that this manuscript is now acceptable for publication, you may indicate that here to bypass the “Comments to the Author” section, enter your conflict of interest statement in the “Confidential to Editor” section, and submit your "Accept" recommendation.

Reviewer #2: All comments have been addressed

2. Is the manuscript technically sound, and do the data support the conclusions?

Reviewer #2: Yes

3. Has the statistical analysis been performed appropriately and rigorously? 

Reviewer #2: Yes

4. Have the authors made all data underlying the findings in their manuscript fully available?

Reviewer #2: Yes

5. Is the manuscript presented in an intelligible fashion and written in standard English?

Reviewer #2: Yes

6. Review Comments to the Author

Reviewer #2: Authors addressed several of my previous comments if not all. However the present data and revised manuscript may be sufficient to convince the audience on the concept and results.

7. PLOS authors have the option to publish the peer review history of their article (what does this mean?). If published, this will include your full peer review and any attached files.

Reviewer #2: No

---

## [Editor Report · Acceptance letter]

PONE-D-25-45571R3

PLOS One

Dear Dr. Kandhavelu,

I'm pleased to inform you that your manuscript has been deemed suitable for publication in PLOS One. Congratulations! Your manuscript is now being handed over to our production team.

Kind regards,

on behalf of

Dr. Suresh Yenugu

Academic Editor

PLOS One